# Comparative Cytogenetic Abnormalities in Paired Choroidal Melanoma Samples Obtained Before and After Proton Beam Irradiation by Transscleral Fine-Needle Aspiration Biopsy and Endoresection

**DOI:** 10.3390/cancers11081173

**Published:** 2019-08-14

**Authors:** Alexandre Matet, Khadija Aït Raïs, Denis Malaise, Martina Angi, Rémi Dendale, Sarah Tick, Livia Lumbroso-Le Rouic, Christine Lévy-Gabriel, Manuel Rodrigues, Gaëlle Pierron, Nathalie Cassoux

**Affiliations:** 1Department of Ophthalmology, Institut Curie, Université Paris Descartes, Sorbonne Paris Cité, F-75005 Paris, France; 2Department of Genetics, Somatic Genetic Unit, Institut Curie, PSL Research University, F-75005 Paris, France; 3Department of Ophthalmology, Institut Curie, PSL Research University, F-75005 Paris, France; 4Department of Radiation Therapy, Institut Curie, PSL Research University, F-75005 Paris, France; 5Quinze-Vingts Ophthalmology National Hospital, F-75012 Paris, France; 6Department of Medical Oncology and INSERM U830, Institut Curie, PSL Research University, F-75005 Paris, France

**Keywords:** uveal melanoma, choroidal melanoma, genomic, chromosome, proton beam therapy, irradiation, fine-needle aspiration biopsy, endoresection

## Abstract

This study compared the cytogenetic profiles of choroidal melanoma samples retrieved before and after proton beam irradiation. Twenty-four consecutive patients who underwent both fine-needle aspiration biopsy (FNAB) during tantalum clip positioning, and endoresection within three months of irradiation, were retrospectively included. Chromosome alterations were explored by array comparative genomic hybridization. Age at diagnosis was 50 ± 14 years, tumor thickness was 8.6 ± 1.7 mm and tumor diameter was 12.4 ± 2.3 mm. Six FNAB samples were non-contributive (25%), versus one endoresection sample (4%) (*p* = 0.049). Among 17 cases with paired contributive samples, the profiles of chromosomes 3 and 8 were identical in all cases, except one with partial chromosome 3 loss on the FNAB sample only. Three cases presented additional discordant aberrations on chromosomes other than 3 or 8q. Overall, we identified monosomy 3 in two cases, 8q gain in six cases, and both alterations in three cases. All cases presented *GNAQ* or *GNA11* mutations assessed by a custom next-generation sequencing panel. Among the six cases with non-contributive initial FNAB, three cases presented abnormal 3 or 8q chromosomes detected on the endoresection material. These results demonstrate the higher rentability of endoresection material for cytogenetic analysis compared to FNAB, and provide clinical evidence of tumor heterogeneity in choroidal melanoma.

## 1. Introduction

Choroidal melanoma, the most frequent intraocular malignancy, leads to metastatic disease in approximately 50% of affected individuals [1]. While the primary tumor can be efficiently managed by either proton beam therapy, plaque brachytherapy, or enucleation for large tumors [2], there is, to date, no effective therapy providing long-term control of metastatic localizations [3,4,5,6]. During tumorigenesis, cells evolving into choroidal melanoma undergo a series of genetic changes altering important cell functions such as chromatin regulation, splicing or translation [7,8]. Apart from modifications in *GNAQ* and/or *GNA11* gene status, frequent and often exclusive mutations occur in *BAP1*, *EIF1AX* or *SF3B1* genes [9]. Moreover, choroidal melanoma cells frequently acquire chromosomal aberrations, that include chromosome 3 monosomy and chromosome 8q gain, and less frequently, partial chromosome 6 additions or deletions [10,11,12,13,14].

In the clinical setting, chromosomal modifications of the primary uveal tumor are of high clinical relevance since they have been extensively shown by numerous investigators to correlate with the metastatic prognosis [9,10,11,12,13,15,16,17]. Therefore, tumor material is retrieved whenever possible at diagnosis in order to determine the metastatic risk, to adapt the timing and modality of the oncological follow-up, and to detect early metastatic disease [18]. For large melanomas requiring primary enucleation, tumor material can be obtained from ocular specimens [10,11]. For small and middle-sized tumors eligible for conservative treatment, tumor material is retrieved by trans-scleral fine-needle aspiration biopsy at the time of tantalum clip positioning prior to irradiation [19], or by transvitreal biopsy performed before or after irradiation, according to local procedures [20].

Moreover, in order to reduce tumor-necrosis-related exudation, retinal ischemia, and to increase the rate of eye preservation, a subset of patients can benefit from endoresection of the tumor scar following proton beam irradiation [21,22]. During this procedure tumor material can also be obtained. However, there is little evidence in the literature regarding the comparative quality of these tumor samples obtained at different timepoints relative to irradiation, thus limiting their relevance regarding metastatic risk prognostication, and their quality for research purposes.

The present study aims at comparing cytogenetic analyses performed on paired choroidal melanoma samples obtained by trans-scleral fine-needle aspiration biopsy and using endoresection material, obtained from single patients before and after proton beam irradiation, respectively.

## 2. Results

### 2.1. Clinical Characteristics

From January 2013 to December 2016, 1388 patients treated at our institution for choroidal melanoma were recorded into the institutional database. Of those, 24 patients underwent both fine-needle aspiration biopsy prior to proton beam therapy, and endoresection less than three months following irradiation, and had cytogenetic analysis performed on both tumor samples, hereby meeting all inclusion criteria.

Mean patient age was 50.2 years (median age 51.0 years). Mean tumor thickness and basal diameter were 8.6 mm and 12.4 mm, respectively. Tumor stage was T2 (*n* = 7) or T3 (*n* = 17) according to the 8th TNM (tumor, node, metastasis) Classification [23]. The mean duration from fine-needle aspiration biopsy to proton therapy was 0.5 month, and the mean duration from proton therapy to endoresection was 1.9 months. The clinical characteristics are detailed in Table 1.

### 2.2. Cytogenetic Analyses

Of the 24 samples retrieved by fine-needle aspiration biopsy, 18 (75%) were contributive whereas 6 (25%) were non-contributive due to insufficient DNA amount or quality. Among the 24 samples collected from the endoresection material, 23 (96%) were contributive and 1 (4%) was non-contributive (*p* = 0.049). Noticeably, for all six patients in whom the fine-needle aspiration biopsy was non-contributive, a contributive sample was obtained from the endoresection material. Moreover, an alteration in chromosome 3 or 8q could be identified in three of these six patients (two cases with 8q addition and 1 case with monosomy 3 and 8q addition). There was no difference between cases with contributive and non-contributive fine-needle aspiration biopsy in terms of tumor thickness (8.7 mm versus 8.4 mm, *p* = 0.47), tumor diameter (12.0 mm versus 13.6 mm, *p* = 0.13) or tumor stage (*p* = 0.63). Five ocular surgeons (N.C., L.D., L.L., C.L. and S.T.) performed fine-needle biopsy sampling. There was no influence of the operator on the efficacy of the sampling for cytogenetic analysis (*p* = 0.59, Chi-squared test).

In total, contributive paired samples were collected from 17 patients (71%). Of these, 16 were concordant regarding alterations in chromosome 3 or 8q, and one case showed discordant samples regarding chromosome 3 status (Case 1). This case presented a partial loss of the long arm subtelomeric region of chromosome 3 on the fine-needle biopsy sample but not on the endoresection sample. Case 1, and two additional cases (Cases 2 and 3), also presented additional aberrations, but affecting chromosomes other than 3 and 8q. These discordances affected chromosomes 4, 6, 9 and 18. In Case 2, an addition of chromosome 4, spanning the short and long arm (4pq+), was detected at the subclonal level. In total, among the 17 paired contributive samples, there were 14 (82%) concordant and 3 (18%) discordant samples, including only one (6%) case presenting a potentially clinically relevant discordance affecting chromosome 3q. The details of all discordant and concordant chromosomal abnormalities detected in the paired samples are summarized in Figure 1.

The comparative cytogenetic profiles of one concordant case (Case 10) and two discordant cases (Cases 1 and 3) are provided in Figure 2, Figure 3 and Figure 4, respectively. The case-by-case detailed clinical characteristics are reported in the Appendix A.

## 3. Discussion

This unique analysis of 24 coupled samples from choroidal melanoma, retrieved before and after irradiation by proton beam therapy, has four main implications:The material retrieved during endoresection has a significantly higher yield, in term of contributive samples, than the biopsies retrieved by fine-needle aspiration;There was a high concordance rate between paired samples, especially regarding chromosomes associated with prognosis (i.e., chromosomes 3 and 8);The discordance observed in a minority of samples may be related to tumor cell content of sample heterogeneity;Cytogenetic alterations are not affected by irradiation, at least within the 2–3 month delay between irradiation and endoresection.

First, the higher rate of contributive samples retrieved by endoresection, as compared to fine-needle aspiration biopsy, is consistent with the lower amount of tumor cells retrieved by fine-needle biopsy. Moreover, it demonstrates that tumor tissue remains exploitable after irradiation, and that despite focal breaks induced by irradiation, the chromosomal architecture of the genomic material is preserved. A previous series from our group identified a similar rate of contributive transscleral biopsies performed before irradiation, with 139 biopsies contributive for cytogenetic analyses, out of 185 (75%) [19]. In a larger cohort of 1103 cases who underwent fine-needle aspiration biopsy, Shields et al. described 1059 contributive biopsies (96%) [16]. Noticeably, these authors employed a different sampling procedure with a 27-gauge needle, and different DNA extraction and processing techniques. Fine-needle aspiration biopsy immediately followed by irradiation (by proton beam or plaque brachytherapy) is a safe procedure in the vast majority of cases, as reported in several large series [14,17,19,24], and recently confirmed in a nationwide cohort study from Denmark [25]. However, extraocular extension to the orbit has been exceptionally described [26,27]. This risk could be further reduced by performing biopsies only after irradiation, and the present study demonstrates that this material would be comparable to pre-irradiation samples in terms of prognostic value.

To the best of our knowledge, this is the largest reported series of paired choroidal melanoma samples obtained before and after irradiation. These results are consistent with two previous case series with a smaller number of samples. Coupland et al. analyzed four coupled samples retrieved by fine-needle aspiration biopsy before Ruthenium^106^ plaque brachytherapy, and after secondary enucleation, that was performed between 7 and 79 months after the primary treatment [28]. These authors used multiplex ligation-dependent probe amplification (MLPA) or microsatellite analysis (MSA), and focused on chromosome 3, 8q, and 6p status. They observed concordant chromosome 3 status among all four cases, concordant chromosome 8 status (among thre cases with pre- and post-irradiation exploration of chromosome 8), and one case with discordant chromosome 6p status (among two cases with exploitable pre- and post-irradiation data). These profiles are similar to those observed in our cohort and originate likely from tumor heterogeneity rather than post-irradiation changes, as discussed below. In a second study, Wackernagel et al. investigated the cytogenetic characteristics of five cases treated by either Ruthenium^106^ plaque brachytherapy (sampled by fine-needle aspiration biopsy, two cases) or Gamma-Knife radiosurgery (sampled by 23-gauge transvitreal biopsy, three cases). They observed a high concordance between the five coupled samples regarding chromosomes 3 and 8q. Noticeably, two cases presented slightly different breakpoints on chromosome 8q before and after radiotherapy. Moreover, the mean delay from irradiation to endoresection was longer than in our study (198 days), varying from 34 days to 526 days.

Taken together, the results from the two above-mentioned studies, and from our study, demonstrate (i) that paired pre- versus post-irradiation samples are highly concordant, (ii) that chromosome 3 and 8 status is rarely modified, and (iii) that minor differences in breakpoint locations may be observed across all chromosomes, and may affect those relevant for prognostication. Given the timing of sampling with respect to irradiation, two phenomena may account for these modifications: either radiation-induced DNA modifications or tumor cellular heterogeneity. Proton particles irradiation result in multiple single and double strand breaks, randomly disseminated over all 46 chromosomes, as ascertained by experimental [29,30] and simulation models [31,32]. DNA modifications between pre- and post-irradiation samples showed a preferential location on certain chromosomes (notably 3, 8q, 6, and 4), and consisted often in additions or deletions of large chromosome segments, not only focal breaks, arguing against a consequence of proton beam irradiation. Moreover, changes at the subclonal level have been identified, such as those presented by the endoresection sample of Case 2 from the present study. Given the highly homogenous radiation pattern administered by proton beam, and the decreasing tumor division rate after treatment, it is unlikely that post-proton therapy changes result in a subclonal organization. Finally, there was no relationship between the degree of discordance and the time elapsed since irradiation, as it should be with post-radiation modifications being more detectable when accumulating cell divisions. In this sense, a recent work by Rodrigues et al., from our institution, showed that metastases occurring in patients treated with proton beam therapy did not show more copy number alterations than in patients treated by enucleation [33] (no patient from this work was included in the present study).

Therefore, these changes are unlikely to result from radiation effects, and are most probably related to intrinsic tumor heterogeneity. The genetic heterogeneity of uveal melanoma has been the subject of extensive investigations, and was identified by several techniques, including chromosome in situ hybridization [34], fluorescence in situ hybridization [35,36], gene expression profile [37,38], and epigenetic studies [39]. It has also been shown to partially correlate with intratumor histologic epithelioid or spindle-cell components [34]. Interestingly, in one study by Augsburger et al., investigators explored discordant gene expression profiles by comparing fine-needle biopsies sampled at two distinct sites in macroscopically heterogenous tumors [37]. While the exact clinical consequences of tumor heterogeneity remain to be determined, the presence of a high-risk subclonal population is sufficient to cause metastatic disease [35], and fine-needle aspiration biopsies carry a certain risk of not detecting a high-risk profile. Furthermore, Rodrigues et al. showed that uveal melanomas accumulate copy number alterations not only during the metastatic process but also during local progression with more copy number alterations being present in larger primary tumors than in smaller ones, suggesting that these aberrations accumulate since early stages [33].

Another phenomenon that may contribute to the discrepancy between fine-needle aspiration and endoresection samples is the low proportion of tumor cells retrieved in some samples (four samples with <60% cellularity are labeled with an asterisk on Figure 1). When analyzing in detail the paired genomic profiles of Case 1, which harbored the lowest estimated percentage of tumor cells at around 30%, the chr11q loss is clearly identifiable (Figure 3). This aberration is present in both samples (identical breakpoints, i.e., filiation within the process), and is observed with a higher dynamic range in the endoresection sample. Therefore, chromosomes 3q and 18q deletions should have been seen with the same log ratio. Of course, the percentage of tumor cells in the sample directly impacts the dynamic of the profile, but regarding the discrepancies between the two samples, it could also be linked to tumor heterogeneity, with two-step alterations for chromosomes 6 and 8 (different breakpoints), and subclones with distinct alterations (chromosomes 3, 9, and 18).

Among aberrations affecting chromosomes other than 3 and 8 detected in this study, the more frequent were chromosome 6 additions (6p) or more rarely deletions (6q), present in 20 of the 24 studied tumors (83%). Chromosome 6 abnormalities are frequent and their prognostic implication has not been to date fully elucidated. Additions of 6p have been long considered to result from alternative genetic tumorigenesis pathways [40], and to be associated with a favorable prognosis [10,41]. Similarly, chromosome 6 alterations were also more frequent in metastatic tumors than in primary tumors according to the article of Rodrigues et al. [33].

This study reports the comparative cytogenetic analysis of paired choroidal melanoma samples all irradiated with the same standardized proton beam modality, and who all underwent endoresection within a narrow timeframe after irradiation in a single center. A novel custom-built NGS panel was also employed to screen all major genetic and chromosomal alterations in a standardized manner. The study presents several limitations, including the selection bias relying on the indication for both fine-needle biopsy and endoresection, restricting the study to T2 and T3 choroidal melanomas. Whether these results can be extrapolated to all melanoma stages is not clear. Moreover, two cytogenetic processing techniques were employed (Agilent and Nimblegen), but they have been previously shown to result in fully comparable profiles [42]). In addition, mean patient age at diagnosis (50.2 years) was lower than the mean age (approximately 60.0 years) observed in large epidemiological studies on uveal melanoma [43]. This selection bias was explained by the more frequent indication of fine-needle aspiration biopsy and endoresection in younger, rather than older, individuals. Indeed, the adapted oncological follow-up with more frequent liver imaging based on the metastatic risk revealed by the fine-needle biopsy is more impactful in younger than older patients. Second, when indicated, endoresection involves a second intraocular surgery, and more frequent follow-up visits, an additional burden more often acceptable for younger patients willing to decrease the long-term rate of post-radiation ocular complications.

Future research perspectives include the fine-tuning adjustment of genetic, clinical, and histopathological predictors for metastatic disease in choroidal melanoma, especially regarding the meaning of other chromosomal aberrations than 3 and 8q. In the past, prognostic analysis has extensively been developed in enucleated eye specimens, that carry a higher clinical metastatic risk due to their high tumor thickness. The advent of fine-needle aspiration biopsy has expanded these investigations to tumors of all volumes, yet with a limited amount of tumor cells. The increasing use of tumor material obtained by endoresection may overcome these limitations on smaller tumors eligible to conservative treatment.

## 4. Materials and Methods

### 4.1. Patients

This study involving human subjects adhered to the tenets of the Declaration of Helsinki and was approved by the Ethics Committee of the Institut Curie (IC 2010-02, approved on 10 February 2011, and IC 2012-03, approved on 11 December 2012). Consecutive patients diagnosed with choroidal melanoma from January 1, 2013 to December 31, 2016, and treated at a tertiary ocular oncology center (Department of Ophthalmology, Institut Curie, Paris, France) were retrospectively identified in the institutional database if they met the following criteria:Conservative management by proton beam irradiation;Fine-needle aspiration biopsy for cytogenetic analysis performed at the time of tantalum clip surgery before irradiation;Endoresection of the tumor scar performed within three months of irradiation, with cytogenetic analysis of the endoresection material.

Each case was staged by its TNM status (tumor, node, metastasis) according to the 8th edition of the American Joint Committee on Cancer Staging manual [23].

### 4.2. Surgical Procedures

#### 4.2.1. Transscleral Fine-Needle Aspiration Biopsy

Fine-needle aspiration biopsy was performed during tantalum clip procedure as previously described [19]. It was indicated when choroidal melanoma thickness was superior to 5 mm, and was performed when tumors did not involve the ciliary body, were not situated too posteriorly to allow surgical access, and were not localized below extraocular muscles. The procedure was performed by four senior ocular oncology specialists (L.L., C.L., N.C. or L.D.). First, the tumor base was localized by transillumination and its borders labeled using patent-blue dye. Then, the biopsy was performed after inserting a 23-gauge needle through the sclera, targeting the tumor center. Tumor material was aspirated with a 2 mL syringe connected to the needle, and immediately purged using Roswell Park Memorial Institute medium into a 1-mL sealed vial. Cyanoacrylate glue was applied onto the sclera at the puncture site to minimize the risk of extraocular dissemination.

#### 4.2.2. Endoresection

Endoresection of the choroidal melanoma scar was indicated and performed within three months of irradiation for choroidal melanomas (not involving the ciliary body), when diameter did not exceed 15 mm and thickness did not exceed 12 mm. The technique was adapted from Damato et al. [44] and Bechrakis et al. [21], as previously described [22]. Briefly, the procedure was performed by a single vitreoretinal surgeon and ocular oncology specialist (N.C.), using 3-port, pars plana vitrectomy with valved 23-gauge trocars on the Constellation device (Alcon, Fort Worth, TX, USA). After central and peripheral vitrectomy, including completion of posterior vitreous detachment, endodiathermy of large choroidal vessels surrounding the tumor base was performed, and 1–2 mL of perfluorocarbon liquid (PFCL) was injected to drain subretinal fluid and prevent subretinal hemorrhage at the posterior pole. Then, endoresection of the tumor was performed. At the beginning of endoresection, the aspiration line of the vitreous cutter was connected to a 2-mL syringe through a 3-port connector, and aspiration of tumor material was performed manually by the assistant surgeon. This syringe was purged into two sealed 1-mL vials sent for cytopathological and cytogenetic analyses. During endoresection, PFCL was re-injected as needed in order to maintain retinal attachment. Throughout the procedure, intraocular pressure was increased to 80–100 mmHg to limit bleeding. The tumor was removed until the inner scleral surface was visible. Cautious endodiathermy was applied around the coloboma after progressively decreasing the intraocular pressure to identify bleeding choroidal and tumor vessels. Under PFCL, endolaser was applied around the coloboma scar, and to the remaining flat tumor residues, combined with cryoapplication of the anterior border, if not accessible to endolaser. Finally, PFCL/silicon oil exchange was performed.

### 4.3. Irradiation Technique

All patients were treated with proton beam therapy following the same irradiation protocol. tantalum clips were localized by computerized tomography (CT-scan) of the orbit. Their position was entered into a virtual three-dimensional reconstruction of the eye and tumor on the EYE-PLAN software (Bebington, Wirral, UK) used for dosimetry calculation. The face of the patient was immobilized into a tailored mask, eyelids were maintained open using a blepharostat, displacing the lacrymal gland superiorly as far away as possible from the incident beam, and the patient was asked to fixate in a precise direction, under the surveillance of an eye tracker. The proton beam was delivered by a Cyclotron device of 73 MeV at a dose of 60 Gray RBE (radiation biologic equivalent), in 4 fractions of 15 Gray RBE each over 4 days. A simulation session was performed on the eve of the first dose. Irradiation was all administered at a single location (Proton Therapy Center of Institut Curie, Orsay, France).

### 4.4. Cytogenetic Analyses

#### 4.4.1. DNA Extraction

Tumor samples were collected in a volume of 10 mL of Hank’s balanced salt solution, at room temperature. They were centrifuged at 2000 RPM, decanted, and pellets were retrieved. They were immediately snap frozen in liquid nitrogen before being stored at −80 °C. DNA was extracted after cells were incubated 2 h with proteinase K, and 1 h with RNAse A, by a phenol/chloroform and Phase Lock Gel light procedure (Eppendorf, Hamburg, Germany).

#### 4.4.2. DNA Quantity and Quality Assessment

Quantity and Quality of DNA Were Estimated Using a 2-Step Procedure. gDNA purity was assessed on a Nanodrop spectrophotometer. For an optimal labeling yield, samples were expected to present A260/A280 and A260/A230 ratios superior to 1.8 and 1.9, respectively. Double-stranded DNA concentration was measured on a Qubit dsDNA BR Assay Kit (Thermo Fisher Scientific, Waltham, MA, USA). On samples presenting low concentrations (10–700 ng), whole genome amplification was performed followed by purification on a QIAamp DNA Mini Kit (Qiagen, Hilden, Germany).

#### 4.4.3. Array Comparative Genomic Hybridization (CGH)

For each sample having sufficient quality and quantity, 700 to 1000 ng of tumor DNA and reference DNA were labeled, purified, and co-hybridized in equal quantity either to the NimbleGen Arrays (Roche, Basel, Switzerland) or Agilent Microarrays (Agilent Technologies, Santa Clara, CA, USA), during 12 to 24 h. Male and female human reference DNA were extracted from human blood for those analyzed on a NimbleGen support, whereas references were provided in the Agilent Kit for those analyzed on an Agilent support. Arrays were washed and scanned according to the technique-specific guidelines.

For samples processed with the Nimblegen technology, images were acquired on a GenePix 4000B scanner with the GenePix V.6.6 software (Molecular Devices, San Jose, CA, USA), and data was extracted using the NimbleScan V.2.5 software. Files produced by the NimbleScan software were then analyzed on SignalMap V.1.9 (Roche, Basel, Switzerland).

For samples processed with the Agilent technology, images were acquired on a SureScan Microarray Scanner using CytoScan software V.2.7, and then analyzed on CytoGenomics software V.3.0.2.11 (Thermo Fisher Scientific, Waltham, MA, USA).

For each sample, the quality of the analysis was evaluated subjectively based on the cytogenetic profile dynamism, the sex mismatch, and the degree of dispersion.

#### 4.4.4. Next-Generation Sequencing (NGS) Panel

A custom-built NGS strategy specifically designed for uveal melanoma (“PUMA”, panel for detection of uveal melanoma alterations) was used to prepare indexed paired-end libraries of tumor DNA using the TruSeq Custom Amplicon Low Input (TSCA-li) library prep kit (Illumina, San Diego, CA, USA). The kit is designed to sequence targeted regions of the genome as large as 40 kb in a single multiplex reaction. We targeted the whole coding DNA sequence spanning 6 genes of interest in choroidal melanoma (*GNAQ*, *GNA11*, *CYSLTR2*, *EIF1AX*, *BAP1,* and *SF3B1*). The design is completed by a low-coverage, pan-genomic backbone of 1536 amplicons distributed along 21 chromosomes (with an over-representation of chromosomes 3, 6, and 8).

Briefly, 10–50 ng of DNA were hybridized to an oligo pool. After removing unbound oligos (washing), an extension/ligation step was conducted, followed by an amplification of the libraries. The libraries were cleaned up, normalized, and pooled. The pool was finally sequenced on a NextSeq mid output of 2 × 100 bp (300 cycles).

#### 4.4.5. Chromosome Abnormalities

The NGS panel was employed to confirm the presence of *GNAQ* or *GNA11* mutations, as expected in uveal melanoma samples. The results of the cytogenetic analyses were categorized as:-Presence of total or partial chromosome 3 loss and/or chromosome 8q gain;-Absence of alteration on chromosomes 3 and 8, but presence of other chromosomal alterations or identification of *GNAQ* and/or *GNA11* mutations on the NGS Panel, confirming the diagnosis of choroidal melanoma;-Non-contributive sample: insufficient DNA amount or quality.

### 4.5. Statistical Analyses

Descriptive and comparative statistics were performed on GraphPad Prism (version 5.0f, GraphPad Software, La Jolla, CA, USA). The Fisher’s exact test and the Mann–Whitney U test were used for contingency and comparative analyses, respectively.

## 5. Conclusions

By comparing the cytogenetic patterns identified on coupled fine-needle aspiration biopsy and endoresection material in 24 patients with choroidal melanoma, we observed a higher rate of contributive samples retrieved by endoresection and a high similarity between the genetic profiles, with marginal or subclonal differences in chromosomal aberrations likely originating from tumor heterogeneity. These results also demonstrated that chromosomal abnormalities remain stable at least until three months post-irradiation, and that tumor material may be confidently sampled either before or after irradiation according to institutional practice patterns.

## Figures and Tables

**Figure 1 cancers-11-01173-f001:**
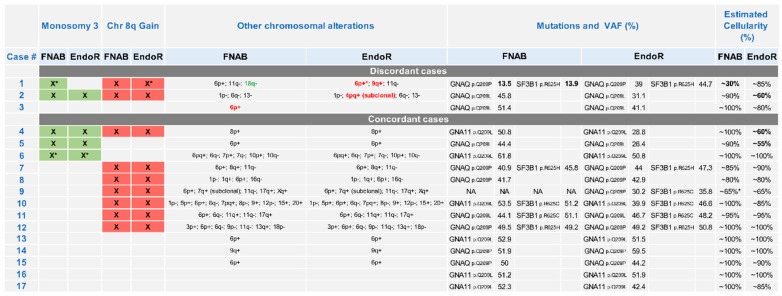
Comprehensive description of molecular profiles observed in paired samples from 24 patients with choroidal melanoma obtained before and after proton beam irradiation by fine-needle aspiration biopsy and endoresection. Presence or absence of ‘Monosomy 3’ or ‘Chromosome 8q gain’, used for risk group stratification, are indicated (green or red box) in the first and second columns. Other chromosomal abnormalities are reported in the third column. Partial alterations are highlighted by an asterisk (*). Among these additional chromosomal gains and losses, those which differed between fine-needle aspiration biopsy (FNAB) and endoresection (EndoR) are labeled in bold red and green, respectively. Mutational status for *GNAQ/GNA11* and *SF3B1* obtained by next-generation sequencing (NGS) are reported in the fourth column: type of mutation in the protein sequence (p.) and frequency of detection (variant allele frequency, VAF). The estimated content of tumoral cells in the samples has been assessed using both comparative genomic hybridization (CGH) array and NGS panel data. There were four samples with cellularity inferior to 60%, which are highlighted in bold.

**Figure 2 cancers-11-01173-f002:**
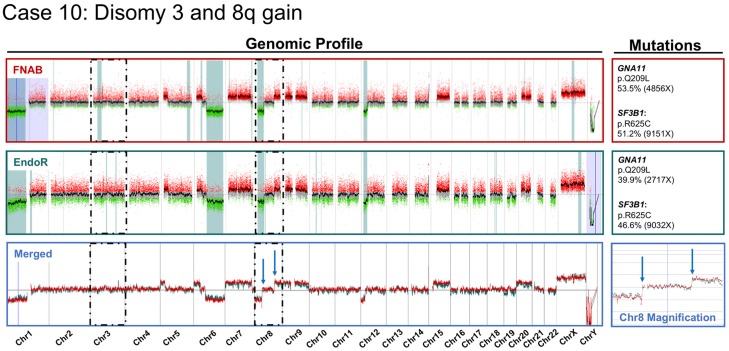
Comparative cytogenetic profiles of Case 10, showing total concordance between the pre-irradiation sample obtained by fine-needle aspiration biopsy (FNAB in dark red) and the post-irradiation endoresection sample (EndoR in dark green). A merged view of superimposed profiles is presented below (light blue) with a magnification on chr8 (chromosome 8) highlighting an example of shared breakpoints. This case also presented the following partial chromosomal aberrations, all similar on both samples: 1p−; 5p+; 6p+; 6q−; 7pq+; and 12p− and whole chr9, 15, and 20 gains. Mutations identified in major genes using a custom-built next-generation sequencing panel for uveal melanoma are reported for each sample (right side of the figure). Tumor DNA was processed with Agilent technology (*y*-axis log2(ratio), *x*-axis genomic position Hg19). Dark dashed boxes focus on chr3 and 8.

**Figure 3 cancers-11-01173-f003:**
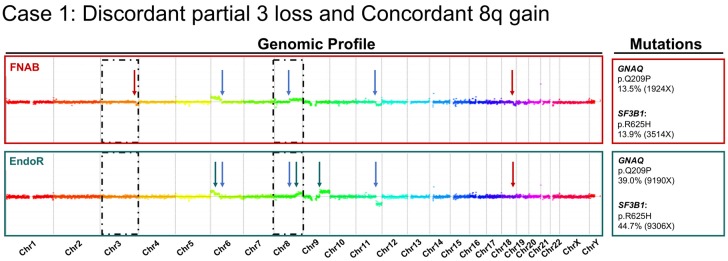
Comparative cytogenetic profiles of Case 1, showing several distinct aberrations between the pre-irradiation sample obtained by fine-needle aspiration biopsy (FNAB in dark red) and the post-irradiation endoresection sample (EndoR in dark green). Shared breakpoints are highlighted by blue arrows (as for 11q−). In addition to a partial chromosome 3q loss on the fine-needle aspiration biopsy sample, this case presented the following partial chromosomal aberrations: 6p+; 18q− (fine-needle aspiration biopsy only); and 9q+; (endoresection only). For 6p+ and 8q+ different breakpoints are present in the endoresection. Mutations identified in major genes using a custom-built next-generation sequencing panel for uveal melanoma are reported for each sample (right side of the figure). Tumor DNA was processed with Nimblegen technology (*y*-axis log2(ratio), *x*-axis genomic position Hg19). Dark dashed boxes focus on chr3 and 8.

**Figure 4 cancers-11-01173-f004:**
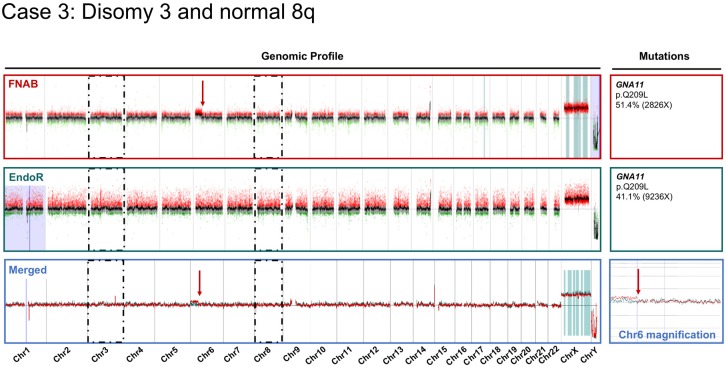
Comparative cytogenetic profiles of Case 3, showing a single discordance between the pre-irradiation sample obtained by fine-needle aspiration biopsy (FNAB in dark red) and the post-irradiation endoresection sample (EndoR in dark green). The magnified area from the merged profiles (blue inlet) displays the breakpoint identified on chromosome 6, present on the fine-needle aspiration biopsy sample only. Mutations identified in major genes using a custom-built next-generation sequencing panel for uveal melanoma are reported for each sample (right side of the figure). Tumor DNA was processed with Agilent technology (*y*-axis log2(ratio), *x*-axis genomic position Hg19). Dark dashed boxes focus on chr3 and 8.

**Table 1 cancers-11-01173-t001:** Clinical characteristics of 24 patients with choroidal melanoma who underwent both fine-needle aspiration biopsy and endoresection before and after proton beam irradiation, respectively.

Gender	Values
Female, No. (%)	11 (46%)
Male, No. (%)	13 (54%)
Age, years	50.2 ± 13.8 (20.8, 73.9)
**Tumor Characteristics at Diagnosis**	
Thickness, mm	8.6 ± 1.7 (4.8, 12.3)
Largest basal diameter, mm	12.4 ± 2.3 (8.3, 15.7)
**Tumor Stage (8th TNM Classification** [23]**)**	
T1, No. (%)	0 (0%)
T2, No. (%)	7 (29%)
T3, No. (%)	17 (71%)
T4, No. (%)	0 (0%)
**Duration between Treatment Events, Month**	
Fine-needle biopsy to proton beam irradiation	0.5 ± 0.1 (0.4, 0.7)
Proton therapy to endoresection	1.9 ± 0.5 (1.3, 3.0)
Fine-needle biopsy to endoresection	2.4 ± 0.5 (1.9, 3.5)
**Follow-up**	
Follow-up after diagnosis, years	3.7 ± 1.6 (0.7, 5.8)
Secondary enucleation, No. (%)	3 (13%)
Metastasis development, No. (%)	3 (13%)
Death, No. (%)	1 (4%)

Quantitative continuous values are provided as mean ± SD (range). TNM—tumor, node, metastasis.

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
