# Peer review of "Comparative Cytogenetic Abnormalities in Paired Choroidal Melanoma Samples Obtained Before and After Proton Beam Irradiation by Transscleral Fine-Needle Aspiration Biopsy and Endoresection"

_cancers, 2019, doi:10.3390/cancers11081173_

Round 1

Reviewer 1 Report

Review of Manuscript ID: cancers-548311

The Comparative cytogenetic abnormalities in paired choroidal melanoma samples obtained before and after proton beam irradiation by transscleral fine needle aspiration biopsy and endoresection by Alexandre MATET at all

 In this paper Matet et al compared the cytogenetic anomalies in FNAB and endoresected material of UM specimens treated by proton-beam irradiation. The paper is well written, the results are summarized concise and the discussion is to the point.

 A few remarks / suggestions.

Title page: aren’t not affiliations 1,3,4,7,8,13 the same? (Also 2 and 10?)

 Results section:

- Line 88: mean age of patients is 50.2 years: seems to be younger compared to other studies? What was the median age and could it be explained by sample bias, or inclusion criteria because of the used treatment protocol? Please discuss this in the discussion.

 -Figure 1: legend is not clear: what is indicated in the yellow box by anomalies in red or green?

- Line 96-100: was there any control for percentage tumour cells in specimen? Alternatively, the results of GNAQ or GNA11 sequencing could be used to give an idea of the amount of tumour cells. Please provide the sequencing results including the observed frequencies of all samples in a table or provide such table as supplementary data and include metastasis and DFS.

- Line 105 was there a relation with the ophthalmologist/operator taking the FNAB ? Was there overlap between the used samples in this report and the Rodrigues (2019, ref 32) paper?

 -Figures 2, 3, 4: these pictures are not sharp! Please provide better quality pictures so that numbers etc are readable

 -Figure 2 and table 1: also gain of chromosomes 7 (except 7qter), 9, 15 and 20 is visible? Why are these gains not listed: is there an explanation in the drop of percentage in GNA11 mutations in the endoresection? Lower percentage of tumour cells?

-Figure 3:The percentage of SF3B1 and GNAQ mutations is very low in the FNAB <14% could this explain the relatively large number of discordant losses and gains?

 -Figure 4: from the provided picture also in the endoresected specimen there is some 6p gain visible and with the lower amount of tumour cells in this sample (41% versus 51%) perhaps using a different threshold setting could solve this discrepancy.

Discussion

-Line 162: The discordance observed in a minority of samples may be related to tumour heterogeneity is to strong since the low percentage of tumour cells could have given a similar effect!

-Line 176-179: Please include the results of the study of Bagger et al Long-Term Metastatic Risk after Biopsy of Posterior Uveal Melanoma (PMID 29705056) in this discussion as they have shown no increased metastatic risk after intraocular tumour biopsy.

- Line 201 the two observed breakpoint differences are (as can be seen in figure 1) 8q and 6p: are these chromosomes less relevant for prognostication??

- Line 286 delete usually (because <3 months is an inclusion criteria)

Author Response

REVIEWER #1

The Comparative cytogenetic abnormalities in paired choroidal melanoma samples obtained before and after proton beam irradiation by transscleral fine needle aspiration biopsy and endoresection by Alexandre MATET et al

In this paper Matet et al compared the cytogenetic anomalies in FNAB and endoresected material of UM specimens treated by proton-beam irradiation. The paper is well written, the results are summarized concise and the discussion is to the point.

A few remarks / suggestions.

Title page: aren’t not affiliations 1,3,4,7,8,13 the same? (Also 2 and 10?)

We thank the Reviewer for pointing out this editing issue. Since the email of each author is required after the affiliation, according to the guidelines to authors, we assumed that each author should be referred to by a different number. We kindly ask the editorial office to let us know whether changes in the affiliation numbering are required.

 Results section:

- Line 88: mean age of patients is 50.2 years: seems to be younger compared to other studies? What was the median age and could it be explained by sample bias, or inclusion criteria because of the used treatment protocol? Please discuss this in the discussion.

Median age was 51.0 years. We agree that mean age of this case series is inferior to the reported mean age of large series, for instance, 60 years in the large epidemiological survey of 2,493 cases by Singh et al. 2003 (Incidence of uveal melanoma in the United States: 1973-1997).

This difference was explained by two factors:

-      First, FNAB was performed more often in younger than elder patients for whom the adapted oncological follow-up based on the metastatic risk was more impactful (consisting in MRI of the liver every 3 months compared to liver ultrasound every 6 months for patient with low, intermediate, or undetermined risk).

-      Second, endoresection was more often performed in younger than elder patients, because it involved a second intraocular surgery, and frequent follow-up visits, for a benefit that was still under evaluation in the first years spanned by this retrospective study. It was therefore more often accepted by younger patients willing to decrease the long-term rate of post-radiation ocular complications.

Both points have been added to the manuscript:

Results, p2 line 92 (line numbers refer to the revised version of the manuscript): “Mean patient age was 50.2 years (median age 51.0 years).”

Discussion, p8 line 433 (§ on Study limitations): “In addition, mean patient age at diagnosis (50.2 years) was lower than the mean age (approximately 60.0 years) observed in large epidemiological studies on uveal melanoma [42]. This selection bias was explained by the more frequent indication of fine-needle aspiration biopsy and endoresection in younger than older individuals. Indeed, the adapted oncological follow-up with more frequent liver imaging based on the metastatic risk revealed by the fine-needle biopsy is more impactful in younger than older patients. Second, when indicated, endoresection involves a second intraocular surgery, and more frequent follow-up visits, an additional burden more often acceptable for younger patients willing to decrease the long-term rate of post-radiation ocular complications.”

 -Figure 1: legend is not clear: what is indicated in the yellow box by anomalies in red or green?

This suggestion helped us to improve our Figure 1. Monosomy 3 and Ch8q gain have been clearly labelled, and other chromosomal alterations are now fully described:

-      identical alterations are reported in plain characters

-      distinct alterations are highlighted in bold characters with colors (green for losses and red for gains)

The Legend to Figure 1 was also revised:

“Figure 1.Comprehensive description of molecular profiles observed in paired samples from 24 patients with choroidal melanoma obtained before and after proton beam irradiation by fine-needle aspiration biopsy and endoresection. Presence or absence of ‘Monosomy 3’ or ‘Chromosome 8q gain’, used for risk groups stratification, are indicated (green or red box) in the first and second columns. Other chromosomal abnormalities are reported in the third column. Partial alterations are highlighted by an asterisk (*). Among these additional chromosomal gains and losses, those which differed between fine-needle aspiration biopsy (FNAB) and endoresection (EndoR) are labelled in bold red and green, respectively. Mutational status for GNAQ/GNA11 and SF3B1 obtained by Next-Generation Sequencing are reported in the fourth column: Type of mutation in p. and frequency of detection (Variant Allele Frequency, VAF). The estimated content of tumoral cells in the samples has been assessed using both Comparative Genomic Hybridization (CGH) array and NGS Panel data. There were four samples with cellularity inferior to 60%, which are highlighted in bold.”

- Line 96-100: was there any control for percentage tumor cells in specimen? Alternatively, the results of GNAQ or GNA11 sequencing could be used to give an idea of the amount of tumour cells. Please provide the sequencing results including the observed frequencies of all samples in a table or provide such table as supplementary data and include metastasis and DFS.

As FNAB samples are small valuable material, mostly dedicated to DNA extraction, they have not been systematically split into two fractions to assess tumor cell content. However, sex mismatch analysis was performed for the CGHa experiments. This method of pairing female gender samples, with male gender DNA reference (and vice versa), allows to have an overall approximation of percentage of tumor cell in the sample by assessing the relative level of gain or loss of chromosome X (if no aberration on this chromosome is observed). As mentionned by the Reviewer, the observed percentage of GNAQ and GNA11 is also a good parameter directly related to this cellularity. This information has been added to Figure1, with an estimation of the tumor cell content based on both CNV dynamic and VAF (variant allele frequency) for mutated genes.

As requested, we have also added a new Supplementary Table, reporting the case-by-case detailed clinical characteristic, including disease-free survival, metastasis and death.

A mention to this Supplementary Table was added to the Text, p4 line 149:

“The case-by-case detailed clinical characteristics are reported in the Supplementary Table 1.

- Line 105 was there a relation with the ophthalmologist/operator taking the FNAB? Was there overlap between the used samples in this report and the Rodrigues (2019, ref 32) paper?

Five onco-ophtalmologists (listed among authors) participated in the FNAB sampling: NC, LD, LL, CL and ST. There was no influence of the operator taking the sample on the efficacy of the sampling for cytogenetic analysis. We performed a contingency analysis with a Chi-square test comparing cases with contributive vs non-contributive FNAB, confirming this observation (P=0.59).

We added the following sentence to the Results:

P3 line 119:“Five ocular surgeons (NC, LD, LL, CL and ST) performed fine-needle biopsy sampling. There was no influence of the operator on the efficacy of the sampling for cytogenetic analysis (P=0.59, Chi-square test).”

There was no overlap between the samples used in this report and the ones from Rodrigues et al. We have added a mention to the manuscript:

P7 lines 387-390:“In this sense, a recent work by Rodrigues et al., from our institution, showed that metastases occurring in patients treated with proton beam therapy did not show more copy number alterations than in patients treated by enucleation [33] (no patient from this work was included in the present study)

 -Figures 2, 3, 4: these pictures are not sharp! Please provide better quality pictures so that numbers etc are readable

We totally agree with the Reviewer and are grateful for this observation. We have reformatted all Figures 2, 3 and 4, to improve their quality and the readability of axis labels. The embedded Figures in the .docx file may also be of slightly lesser resolution than the original files.

 -Figure 2 and table 1: also gain of chromosomes 7 (except 7qter), 9, 15 and 20 is visible? Why are these gains not listed: is there an explanation in the drop of percentage in GNA11 mutations in the endoresection? Lower percentage of tumour cells?

We thank the Reviewer for this observation. We had initially focused on segmental aberrations, so whole chromosomes gains and/or losses were not reported. As recommended, we have provided a new table in which all alterations are listed, including those affecting the above-mentioned chromosomes. The Figure 1 has also been reformatted with the inclusion of GNAQ/11 percentages and tumor cell content estimation, and the color code has been simplified.

Regarding Case #11 (renamed Case #10 in this revision due to a classification change of Case #6): As the reviewer pointed out, the tumoral cell content is lower for the second sample: around 5% less, which is coherent for SF3B1 status. Moreover the DNA quality of the endoresection sample is poorer than the one obtained from the FNAB, as observed on the CGHa profile. In order to assess this second point, we carefully reanalyzed the bam file of the NGS panel analysis. There was more noise on the analysis of the endoresection sample for the GNA11 position, and some reads have been filtered out during the QC process. It appears that this filtering step led to a small imbalance between normal and mutated alleles, which can explain this 5% additional discrepancy between the VAF of SF3B1 and GNA11 in this case.

We have modified Figure 2 and the corresponding Legend:

 “Comparative cytogenetic profiles of Case #11, showing total concordance between the pre-irradiation sample obtained by fine-needle aspiration biopsy (FNAB in dark red) and the post-irradiation endoresection sample (EndoR in dark green). A merged view of superimposed profiles is presented below (light blue) with a zoom on chr8 highlighting an exemple of commun breakpoints. This case also presented the following partial chromosomal aberrations, all similar on both samples: 1p-; 5p+; 6p+; 6q-; 7pq+ and 12p- and whole chr9, 15 and 20 gains. Mutations identified in major genes using a custom-built next-generation sequencing panel for uveal melanoma are reported for each sample (right side of the Figure). Tumor DNA was processed with the Agilent technology. (Y axis Log2(Ratio), X axis genomic position Hg19). Dark dashed boxes focus on chr3 and 8.”

-Figure 3: The percentage of SF3B1 and GNAQ mutations is very low in the FNAB <14% could this explain the relatively large number of discordant losses and gains?

For Case 1 illustrated on Figure 3, as the reviewer has noticed, the cellularity of the FNAB sample is the lowest of the case series. Even if the estimated percentage of tumor cells is around 30%, the chr11q loss is clearly identifiable. This aberration is present in both samples (identical breakpoints, i.e. filiation within the process), and is observed with a higher dynamic range in the endoresection sample. So chr3q and chr18q deletions should have been seen with the same log ratio. Of course the percentage of tumor cells in the sample directly impacts the dynamic of the profile, but regarding the discrepancies between the two samples, it could also been linked to tumor heterogeneity, with two steps alterations for chromosomes 6 and 8 (different breakpoints) and subclones with distinct alterations for chr3, chr9 and chr18.

Moreover to exclude any doubt, we performed a genotyping analysis (using all polymporphisms included in our NGS panel, and also confirmed this filiation with an independent technic using Authentifiler kit Life Tech) and we assessed that the two samples belong to the same patient.

We have added the following paragraph to the Discussion:

P7 line 382: “Another phenomenon thay may contribute to the discrepancy between fine-needle aspiration and endoresection samples is the low proportion of tumor cells retrieved in some samples (four samples with <60% cellularity are labelled with an asterisk on Figure 1). When analyzing in detail the paired genomic profiles of Case 1, which harbored the lowest estimated percentage of tumor cells at around 30%, the chr11q loss is clearly identifiable (Figure 3). This aberration is present in both samples (identical breakpoints, i.e. filiation within the process), and is observed with a higher dynamic range in the endoresection sample. So chromosomes 3q and 18q deletions should have been seen with the same log ratio. Of course the percentage of tumor cells in the sample directly impacts the dynamic of the profile, but regarding the discrepancies between the two samples, it could also been linked to tumor heterogeneity, with two-step alterations for chromosomes 6 and 8 (different breakpoints), and subclones with distinct alterations (chromosomes 3, 9 and 18).”

 -Figure 4: from the provided picture also in the endoresected specimen there is some 6p gain visible and with the lower amount of tumour cells in this sample (41% versus 51%) perhaps using a different threshold setting could solve this discrepancy.

As suggested by the Reviewer, we have tried to reanalyze the data applying different segmentation algorithms, but whatever the method and threshold employed, we did not evidence any breakpoint on chr6p for the endoresection sample. DNA of Case 3 used to perform both CGHa profiles and sequencing of GNAQ gene shows a hot spot mutation % of 51% and 41% in the FNAB and the endoresection samples, respectively, fitting perfectly with “high content of tumoral cells” samples category. The chr6q gain has a low level of dynamic range regarding the 100% of cellularity. One hypothesis can be that 2 subclonal populations (both mutated for GNAQ) co-exist in the tumor one with the chr6p gain, and one without (comparison can be made with the dynamic of chrX). The tumoral cell content of the endoresection sample is estimated around 80%. Of course, this 10% difference can have “diluted” the aberration, or the sample can also have been enriched with the subclone where chr6 gain was absent. 

To note, Case #11 displayed on Figure 4 has been renamed Case #10 due to a change in Case #6 classification.

Discussion

-Line 162: The discordance observed in a minority of samples may be related to tumour heterogeneity is too strong since the low percentage of tumour cells could have given a similar effect!

This has been corrected as (line 331): “The discordance observed in a minority of samples may be related to tumor cell content of sample heterogeneity “

-Line 176-179: Please include the results of the study of Bagger et al Long-Term Metastatic Risk after Biopsy of Posterior Uveal Melanoma (PMID 29705056) in this discussion as they have shown no increased metastatic risk after intraocular tumour biopsy.

We thank the Reviewer for pointing out this relevant reference that we have now included in the Discussion, as follows:

P7 line 346:“Fine-needle aspiration biopsy immediately followed by irradiation (by proton beam or plaque brachytherapy), is a safe procedure in the vast majority of cases, as reported in several large series [14,17,19,24], and recently confirmed in a nationwide cohort study from Denmark [25].

- Line 201 the two observed breakpoint differences are (as can be seen in figure 1) 8q and 6p: are these chromosomes less relevant for prognostication??

We agree that this statement should be modulated since chromosome 8q gain is of prognostic importance, and there is no definitive certainty regarding the prognostic value of chromosome 6p gain. We have therefore rephrased as follows:

P7 line 373: “(iii) that minor differences in breakpoint locations may be observed across all chromosomes, and may affect those relevant for prognostication.

- Line 286 delete usually (because <3 months is an inclusion criteria)

We have modified the text accordingly:

P8 line 479:“… and performed within 3 months of irradiation”

Points that must be improved in the Reviewer Checklist:Presentation of the Results

We believe that the changes made in response to the above-mentioned comments, and the requested improvement brought to Figures 1 to 4, has improved the clarity of the Results.

Miscellaneous changes:

Reanalysis of the genomic profile of Case #6 showed that it did not present a partial chromosome 3 loss but only a partial chromosome 3 gain. For graphical purposes (on Figure 1), it has been renumbered as Case #13, and all cases from 7 to 13 renumbered from 6 to 12). Changes in the Text and Figure legends have been made where necessary.

Reviewer 2 Report

Uveal melanoma is the deadest ocular tumor with a 50% death rate, there is  no effective treatment to reduce the death rate in the clinic.  Pathological classification based on biopsy is essential for prognosis of patients. Proton beam irradiation is one of common localized treatments to relieve the tumor burden without significant change of tumor heterogeneity.  This is a well-written manuscript with reliable data and clear conclusion. Although the idea to compare pathological profiles from biopsy sample and endorsectioned tumor and the concordance result are not new, the additional evidence provide the community more convinced picture on the biopsy sampling. 

Author Response

REVIEWER #2

Uveal melanoma is the deadest ocular tumor with a 50% death rate, there is no effective treatment to reduce the death rate in the clinic.  Pathological classification based on biopsy is essential for prognosis of patients. Proton beam irradiation is one of common localized treatments to relieve the tumor burden without significant change of tumor heterogeneity. This is a well-written manuscript with reliable data and clear conclusion. Although the idea to compare pathological profiles from biopsy sample and endorsectioned tumor and the concordance result are not new, the additional evidence provide the community more convinced picture on the biopsy sampling. 

We are grateful to Reviewer #2 for these encouraging comments. 

Reviewer 3 Report

The research article titled "Comparative cytogenetic abnormalities in paired choroidal melanoma samples obtained before and after proton beam irradiation by transscleral fine-needle aspiration biopsy and endoresection" by Matet et al., is a very interesting article.

The subject matter is very appealing to the scientific community, experiments are well-designed and considering the high sample size the results seem convincing.  The conclusions of the study are being supported by scientific evidence and the authors have used adequate and appropriate references.  Comparison between cytogenetic analysis and FNAB is a very scientifically sound approach to provide clinical evidence of tumor heterogeneity in choroidal melanoma. This will help clinicians in choosing appropriate treatment modalities for choroidal melanoma.

Author Response

REVIEWER #3

The research article titled "Comparative cytogenetic abnormalities in paired choroidal melanoma samples obtained before and after proton beam irradiation by transscleral fine-needle aspiration biopsy and endoresection" by Matet et al., is a very interesting article.

The subject matter is very appealing to the scientific community, experiments are well-designed and considering the high sample size the results seem convincing. The conclusions of the study are being supported by scientific evidence and the authors have used adequate and appropriate references. Comparison between cytogenetic analysis and FNAB is a very scientifically sound approach to provide clinical evidence of tumor heterogeneity in choroidal melanoma. This will help clinicians in choosing appropriate treatment modalities for choroidal melanoma.

We are grateful to Reviewer #3 for these constructive comments.

Round 2

Reviewer 1 Report

thank you for your adequate anwers

a few typo's

line 268: thay --> that

line 298: delete - than older- or rephrase

line 373: irradiationa --> irradiations